



# Analysis of Control-Oriented Wake Modeling Tools Using Lidar Field Results

Jennifer Annoni[1], Paul Fleming[1], Andrew Scholbrock[1], Jason Roadman[1], Scott Dana[1],
Christiane Adcock[1], Fernando Porte-Agel[2], Steffen Raach[3], Florian Haizmann[3], and David Schlipf[3]

[1]National Wind Technology Center, National Renewable Energy Laboratory, Golden, CO, 80401, USA
[2]Ecole Polytechnique Federale De Lausanne (EPFL), Lausanne, Switzerland
[3]Stuttgart Wind Energy (SWE), University of Stuttgart, Allmandring 5B, 70569 Stuttgart, Germany

*Correspondence to:* Jennifer Annoni (jennifer.annoni@nrel.gov)

**Abstract.**

Wind turbines in a wind farm operate individually to maximize their own performance regardless of the impact of aerodynamic interactions on neighboring turbines. Wind farm controls can be used to increase power production or reduce overall structural loads by properly coordinating turbines. One wind farm control strategy that is addressed in literature is known as

wake steering, wherein upstream turbines operate in yaw misaligned conditions to redirect their wakes away from downstream turbines. The National Renewable Energy Laboratory (NREL) in Golden, CO conducted a demonstration of wake steering on a single utility-scale turbine. In this campaign, the turbine was operated at various yaw misalignment setpoints while a lidar mounted on the nacelle scanned five downstream distances. The lidar measurements were combined with turbine data, as well as measurements of the inflow made by a highly instrumented meteorological mast upstream. The full-scale measurements are

used to validate controls-oriented tools, including wind turbine wake models, used for wind farm controls and optimization. This paper presents a quantitative comparison of the lidar data and controls-oriented wake models under different atmospheric conditions and turbine operation. The results show good agreement between the lidar data and the models under these different conditions.

## 1 Introduction

Control strategies can be used to maximize power production of a wind farm, reduce structural loads to increase the lifetime of turbines in a wind farm, and better integrate wind energy into the energy market (Johnson and Thomas (2009); Boersma et al. (2017)). Typically, wind turbines in a wind farm operate individually to maximize their own performance regardless of the impact of aerodynamic interactions on neighboring turbines. As a result, there is the potential to increase power and reduce overall structural loads by properly coordinating turbine control actions. Two common wind farm control strategies in

literature include wake steering and axial induction control. There has been a significant amount work done on wake steering in simulation, showing that this method has potential to increase power production (Gebraad et al. (2016)). Wake steering typically uses yaw misalignment of the turbines to redirect the wake around downstream turbines. Various computational fluid dynamics (CFD) simulations and wind tunnel experiments have shown that this method can increase power without



substantially increasing turbine loads (Gebraad et al. (2016); Fleming et al. (2014); Jiménez et al. (2010)). Yaw-based wake steering control has also been used in optimization studies of turbine layouts to improve the annual energy production of a wind farm (Fleming et al. (2016); Thomas et al. (2015); Stanley et al. (2017)). Recent CFD studies have determined that the shape of the wake and atmospheric stability are significant factors in wake steering (Vollmer et al. (2016)).

Control-oriented models are essential for developing and deploying wake steering strategies in wind farms. In particular, control-oriented models can be used in open-loop control where a look-up table is generated a priori and used in the field. Alternatively, because of its computational efficiency, a control-oriented model can be used to perform online optimization with feedback to adjust to changing conditions in the atmosphere or wind farm, (e.g. turbine down for maintenance). Lastly, control-oriented models are also useful for large-scale analysis and assessing the impact of controls and optimization on annual

energy production. Overall, these models are critical to the success of wind farm controllers and as a result, full-scale validation of these control-oriented models is essential and a high priority in this area of research.

A full-scale demonstration of wake steering is necessary to understand its advantages and to validate the benefits predicted by simulations. Wind tunnel tests have been conducted that show encouraging results that match simulation results based on wake redirection (Campagnolo et al. (2016); Schottler et al. (2016)). In addition, there are preliminary results of the benefits

of wake steering from an offshore commercial wind farm (Fleming et al. (2017c)). NREL also conducted a detailed full-scale demonstration in which a utility-scale turbine was operated at various yaw offsets while the wake was measured using a scanning lidar. In this paper, the lidar data collected from this campaign are used to validate controls-oriented tools that are used for wind farm control. The main contributions of this work include a review of control-oriented models used for wake steering as well as a quantitatively comparing these models and full-scale lidar results. The results between the wake models

and lidar data show good agreement under various atmospheric and turbine operating conditions, as shown in Section 4. This is an encouraging result that provides confidence in previously reported benefits of wake steering. The wind farm control tools, including wake models, turbine models, and lidar models, used in wind farm controls are introduced in Section 2. The field campaign is briefly introduced in Section 3. Finally, Section 5 provides conclusions and future work.

## 2   Modeling

FLOw Redirection and Induction in Steady State (FLORIS) is defined as a set of controls and optimization tools used for wind farm control developed at NREL and TU Delft (see Figure 1). This tool models the turbine interactions in a wind farm and can be used to perform real-time optimizations to improve wind farm performance and integrate supervisory and data acquisition data (SCADA) data collected at wind farms. This section focuses on the wake models, turbine models, and the lidar model used in this paper.

### 2.1   Wake Model

The wake models available in FLORIS include the Jensen model (Jensen (1983)), the multizone wake model (Gebraad et al. (2016)), and the self-similar wake model with contributions from (Bastankhah and Porté-Agel (2014, 2016); Abkar and Porté-



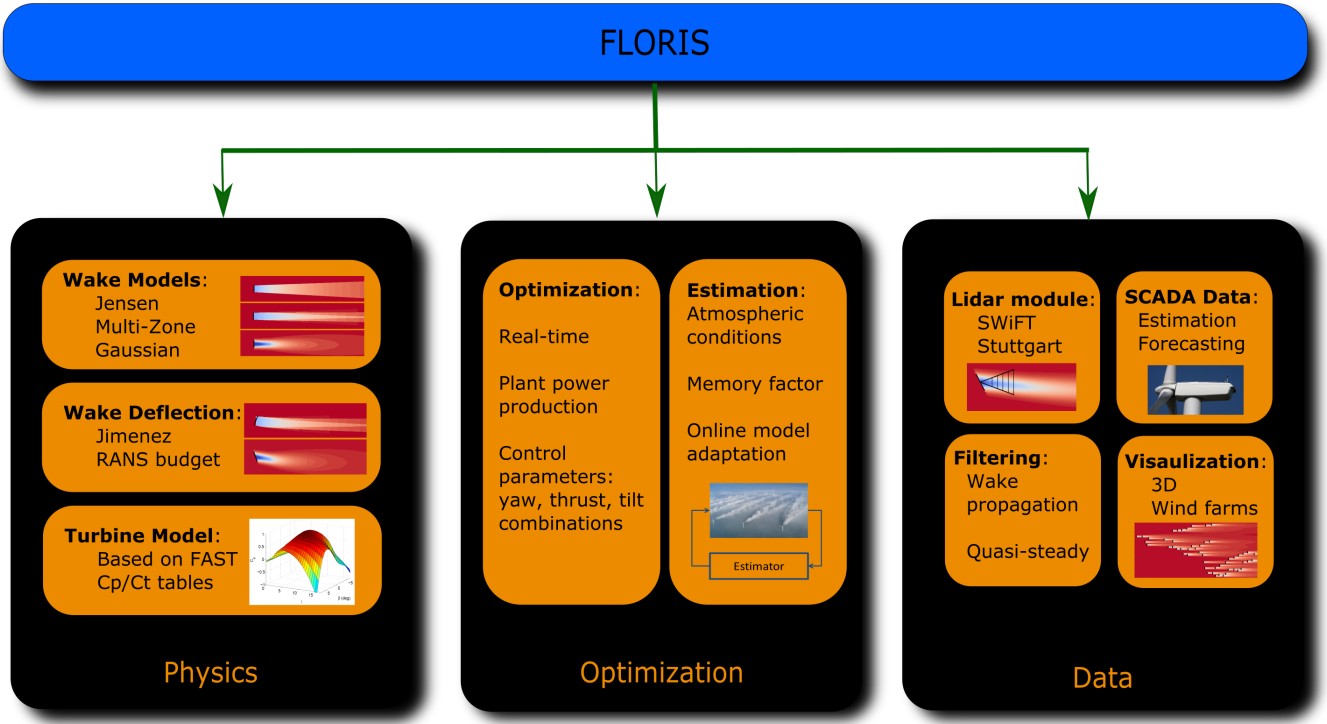

**Figure 1.** The FLORIS toolset is comprised of three main sections: (1) physics, (2) optimization, and (3) data.

Agel (2015); Niayifar and Porté-Agel (2016); Dilip and Porté-Agel (2017)). Although only these three wake models are addressed and implemented in FLORIS, any wake model can be substituted into the FLORIS framework for real-time optimization of a wind farm. This paper also demonstrates the modular framework for FLORIS and will address the benefits of adding complexity to wake models used to characterize the aerodynamic interactions between turbines.

### 2.1.1 Jensen Model

The Jensen model has been used for numerous studies on wind farm controls (Jensen (1983); Johnson et al. (2006); Katic et al. (1986)). This model has a low computational cost because of its simplicity and is based on assumptions that there is a steady inflow and linear wake expansion, and that the velocity in the wake is uniform at a cross section downstream. The turbine is modeled as an actuator disk with uniform axial loading in a steady uniform flow.

10    Consider the example of a turbine operating in free stream velocity, $U_\infty$. The diameter of the turbine rotor is denoted by $D$ and the turbine is assumed to be operating at an induction factor, $a$. A cylindrical coordinate system is placed at the rotor hub of the first turbine with the streamwise and radial distances denoted by $x$ and $r$, respectively. The velocity profile at a location $(x, r)$ is computed as:

$$u(x, r, a) = U_\infty(1 - \delta u(x, r, a)) \tag{1}$$





where the velocity deficit, $\delta u$, is given by:

$$\delta u = \begin{cases} 2a \left( \frac{D}{D+2kx} \right)^2, & \text{if } r \leq \frac{D+2kx}{2}. \\ 0, & \text{otherwise.} \end{cases} \tag{2}$$

In this model, the velocity, $u$, is defined in the axial ($x$) direction and the remaining velocity components are neglected. The wake is parameterized by a tuneable nondimensional wake decay constant, $k$. Typical values of $k$ range from 0.01 to 0.5 depending on ambient turbulence, topographical effects, and turbine operation. For example, if the ambient turbulence is high, then the wakes within the wind farm will recover faster because of the mixing with the atmospheric flow. As a result, the $k$ value will be higher indicating that the wake will recover faster. There is no standard numerical rule for how $k$ varies with turbulence intensity.

**Limitations:** The Jensen model can be used to compute the power production and velocity deficit of a turbine array. This model has been used to determine operating conditions of a wind farm to maximize power (Johnson and Thomas (2009)). However, the model neglects the effects of added turbulence in the downstream wake as a result of varying turbine operation. The model assumptions are based on a steady inflow acting on an actuator disk with uniform axial loading. Despite its limitations, the Jensen model can be computed in fractions of a second and can provide some insight into turbine interaction in a short amount of time. In addition, if uncertainty is included, the Jensen model performs well and successfully predicts wake interactions under normal operating conditions.

### 2.1.2 Multizone Model

The multizone model, developed in (Gebraad and Van Wingerden (2014)), is a modification of the Jensen model described in the previous section. Modifications were made to Jensen model the wake velocity profile and effects of partial wake overlap, especially in yawed conditions. The multizone model defines three wake zones, $q$: (1) near-wake zone, (2) far-wake zone, and (3) mixing-wake zone. The effective velocity at the downstream turbine, $i$, is found by combining the effects of each of the wake zones of the upstream turbine, $j$:

$$u_i = U_\infty \left( 1 - 2 \sqrt{ \sum_j \left[ a_j \sigma_{q=1}^3 c_{j,q} (X_i \min(\frac{A_{j,i,q}^{\text{overlap}}}{A_i}, 1)) \right]^2 } \right) \tag{3}$$

where $X_i$ is the $x$ location of turbine $i$, $A_{j,i,q}^{\text{overlap}}$ is the overlap area of a wake zone, $q$ of a turbine $i$ with the rotor of turbine $j$, and $c_{i,q}(x)$ defines the recovery of a zone $q$ to the free stream conditions:

$$c_{i,q}(x) = \left( \frac{D_i}{D_i + 2k_e m_{U,q}(\gamma_i)[x - X_i]} \right)^2 \tag{4}$$

where $m_{U,q}$ is defined as:

$$m_{U,q}(\gamma_i) = \frac{M_{U,q}}{\cos(a_U + b_U \gamma_i)} \tag{5}$$



where $q = 1, 2, 3$ corresponding to the three wake overlap zones, where $a_U$ and $b_U$ are tuned model parameters, $D_i$ is the rotor diameter of turbine $i$, $\gamma_i$ is the yaw offset of turbine $i$, and $M_{U,q}$ are tuned scaling factors that ensure that the velocity in the outer zones of the wake will recover to the free stream conditions faster than the inner zone. The parameters of the model were tuned to match the results from high-fidelity wake simulations (Gebraad et al. (2016)) . The most influential parameter is $k_e$,

because it defines both wake expansion and wake recovery. Additional details can be found in (Gebraad et al. (2016)).

**Limitations:** The multizone model was tuned with high-fidelity models, to characterize turbine interactions when turbines were operating in partial wake or yawed conditions. The multizone model is a computationally inexpensive model that in suitable for online optimization studies to improve wind farm performance. However, there are ten free parameters that can be tuned in this model and the tuning can be sensitive depending on the parameters chosen to tune. Like the Jensen model, this

model does not have any sensitivity to turbulence intensity or added turbulence generated by an upstream turbine.

### 2.1.3 Gaussian Model

Lastly, a Gaussian model is incorporated in the overall FLORIS wake modeling and controls tools. This model was introduced by several recent papers including (Abkar and Porté-Agel (2015); Bastankhah and Porté-Agel (2014, 2016); Niayifar and Porté-Agel (2016); Dilip and Porté-Agel (2017)). This model includes a Gaussian wake, in the $y$ and $z$ directions, to describe

the velocity deficit, added turbulence based on turbine operation, and atmospheric stability.

**Velocity Deficit:** The velocity deficit of a wake is computed using a Gaussian wake based on self-similarity theory often used in free shear flows, (Pope (2000)). An analytical expression for the three-dimensional velocity deficit behind a turbine in the far wake can be derived from the simplified Navier-Stokes equations as:

$$\frac{u(x,y,z)}{U_\infty} = 1 - Ce^{-(y-\delta)^2/2\sigma_y^2}e^{-(z-z_h)^2/2\sigma_z^2} \tag{6}$$


$$C = 1 - \sqrt{1 - \frac{(\sigma_{y0}\sigma_{z0})M_0}{\sigma_y\sigma_z}}$$

$$M_0 = C_0(2 - C_0)$$

$$C_0 = 1 - \sqrt{1 - C_T}$$

where $C$ is the velocity deficit at the wake center, $\delta$ is the wake deflection (see Section 2.2), $z_h$ is the hub height of the turbine, $\sigma_y$ defines the wake width in the $y$ direction, and $\sigma_z$ defines the wake width in the $z$ direction. Each of these parameters are defined with respect to turbine $i$; subscripts are excluded for brevity. The subscript "0" refers to the initial values at the start of the far wake, which is dependent on ambient turbulence intensity, $I_0$, and the thrust coefficient, $C_T$. For additional details

on near-wake calculations, refer to (Bastankhah and Porté-Agel (2016)). Abkar and Porté-Agel (2015) demonstrate that $\sigma_y$ and $\sigma_z$ grow at different rates based on lateral wake meandering ($y$-direction) and vertical wake meandering ($z$-direction). The velocity distributions $\sigma_z$ and $\sigma_y$ are defined as:

$$\frac{\sigma_z}{d} = k_z\frac{(x - x_0)}{d} + \frac{\sigma_{z0}}{d}, \qquad \text{where } \frac{\sigma_{z0}}{d} = \frac{1}{2}\sqrt{\frac{u_R}{u_\infty + u_0}} \tag{7}$$





$$\frac{\sigma_y}{d} = k_y \frac{(x - x_0)}{d} + \frac{\sigma_{y0}}{d}, \qquad \text{where } \frac{\sigma_{y0}}{d} = \frac{\sigma_{z0}}{d} \cos\gamma \qquad (8)$$

where $k_y$ defines the wake expansion in the lateral direction and $k_z$ defines the wake expansion in the vertical direction. For this study, $k_y$ and $k_z$ are set to be equal and the wake expands at the same rate in the lateral and vertical directions. The wakes

are combined using the traditional sum of squares method (Katic et al. (1986)), although alternate methods are proposed in Niayifar and Porté-Agel (2016).

**Atmospheric Conditions:** This model also accounts for physical atmospheric quantities such as shear, veer, and changes in turbulence intensity (Abkar and Porté-Agel (2015); Niayifar and Porté-Agel (2016)). Shear, veer, and turbulence intensity measurements are typically available in field measurements and will be used to characterize atmospheric conditions in this

particular model. These are physical parameters that can be available in field measurements. It should be noted that these three parameters do not sufficiently characterize atmospheric stability as defined in Stull (2012). Other parameters, such as vertical flux and temperature profiles, are necessary to fully capture atmospheric stability.

The Gaussian model is a three-dimensional wake model that includes shear by using the power log law of wind:

$$\frac{u}{U_\infty} = \left( \frac{z}{z_{hub}} \right)^{\alpha_s} \qquad (9)$$

where $\alpha_s$ is the shear coefficient. A high shear coefficient, $\alpha_s > 0.2$, is typically used for stable conditions and a low shear coefficient, $\alpha_s < 0.2$, is typically used for unstable conditions (Stull (2012)).

This wake model also takes into account veer associated with wind direction change across the rotor. A rotation factor is added to the Gaussian wake (6) such that:

$$\frac{u(x,y,z)}{U_\infty} = 1 - C e^{-(a(y-\delta)^2 - 2b(y-\delta)(z-z_{hub}) + c(z-z_{hub})^2)} \qquad (10)$$


$$a = \frac{\cos^2\phi}{2\sigma_y^2} + \frac{\sin^2\phi}{2\sigma_z^2}$$

$$b = -\frac{\sin 2\phi}{4\sigma_y^2} + \frac{\sin 2\phi}{4\sigma_z^2}$$

$$c = \frac{\sin^2\phi}{2\sigma_y^2} + \frac{\cos^2\phi}{2\sigma_z^2}$$

where $\phi$ is the amount of veer, i.e. change in wind direction, across the rotor. This equation represents a standard Gaussian rotation.

Lastly, turbulence intensity is accounted for in the model by linking ambient turbulence intensity to wake expansion. An empirical relationship is provided in Niayifar and Porté-Agel (2016):

$$k_y = 0.38371 I + 0.003678 \qquad (11)$$

where $I$ represents the streamwise turbulence intensity. As stated previously, $k_y$ and $k_z$ are considered equal in this study.



**Added Turbulence:** This wake model also computes added turbulence generated by turbine operation and ambient turbulence conditions. For example, if a turbine is operating at a higher thrust, this will cause the wake to recover faster. Conversely, if a turbine is operating at a lower thrust, this will cause the wake to recover slower. Conventional linear flow models have a single wake expansion parameter that does not change under various turbine operating conditions. Niayifar and Porté-Agel (2016) provided a model that incorporated added turbulence caused by turbine operation:

$$I = \sqrt{\sum_{j=0}^{N}(I_j^+)^2 + I_0^2} \tag{12}$$

where $N$ is the number of turbines influencing the downstream turbines, $I_0$ is the ambient turbulence intensity, and the added turbulence caused by turbine $i$, $I_i^+$ is computed as:

$$I^+ = A_{overlap}\left(0.8a_i^{0.73}I_0^{0.35}(x/D_i)^{-0.32}\right) \tag{13}$$

where $I_0$ is the ambient turbuence intensity and $a$ is the axial induction factor of the turbine, which can be defined in terms of $C_T$ based on Burton et al. (2001); Bastankhah and Porté-Agel (2016):

$$a \approx \frac{1}{2\cos\gamma}\left(1 - \sqrt{1 - C_T\cos\gamma}\right)$$

In Niayifar and Porté-Agel (2016), the number of turbines, $N$, used to determine the added turbulence is $N = 1$ where the one turbine is the turbine with the strongest wake effect on the turbine being evaluated. In the formulation used in FLORIS, $N$ is determined by the number of turbines a pre-defined distance to the downstream turbine rather than only including the influence of one turbine. For example, the model included in FLORIS arbitrarily considers contributions to the added turbulence intensity from turbines within 15D. This has shown to be beneficial, especially with closely spaced turbines. Studies have shown that the added turbulence intensity has reached an equilibrium point between two and three turbines downstream (Chamorro and Porté-Agel (2011)).

**Capabilities and Limitations:** The Gaussian model is an analytical model with approximations made from the simplified Navier-Stokes equations based on free shear flows. However, it relies on a linear wake expansion model and has six tuning parameters based on empirical relationships for wake expansion and turbulence intensity (Equations 11 and 13). The main benefits of this model come from the ties to physical measurements in the field such as shear, veer, and turbulence intensity and its roots in self-similar free shear flow theory.

## 2.2 Wake Deflection

The wake models defined earlier include wake deflection models that approximate the amount of lateral movement based on yaw misalignment of the turbine. It is important to note that when a turbine is positively misaligned, the turbine has rotated counter-clockwise to be misaligned with the oncoming wind (conventional right-hand-rule). Two wake deflection models are defined in the FLORIS wind farm modeling and controls framework and are briefly described in this section.





### 2.2.1 Jimenez Model

An empirical formulation was presented in Jiménez et al. (2010) and used in the multizone formulation (Gebraad et al. (2016)). When a turbine is yawed, it exerts a force on the flow that causes the wake to deflect and deform in a particular direction. The angle at the wake centerline is defined as:

$$\xi(x) \approx \frac{\xi_{init}^2}{1 + 2k_d \frac{x}{D}} \tag{14}$$

$$\xi_{init}(a, \gamma) = \frac{1}{2} \cos^2 \gamma \sin \gamma C_T$$

where $\xi_{init}$ is the initial skew angle from the wake centerline and $k_d$ is a tuneable deflection parameter. In Gebraad et al. (2016), the wake deflection angle is integrated to determine the amount of deflection, and $\delta$ achieved by yaw misalignment in the spanwise ($y$ direction):

$$\delta(x) = \int_0^x \tan \xi(x) dx \tag{15}$$

$$\delta(x) \approx \frac{\xi_{init} \left( 15 \left( \frac{2k_d x}{D} + 1 \right)^4 + \xi_{init}^2 \right)}{\frac{30k_d}{D} \left( \frac{2k_d x}{D} + 1 \right)^5} - \frac{\xi_{init} D \left( 15 + \xi_{init}^2 \right)}{30k_d}$$

The deflection, $\delta$, is achieved by integrating a second-order Taylor series approximation shown in Gebraad et al. (2016).

### 2.2.2 Bastankhah and Porte-Agel Model

In Bastankhah and Porté-Agel (2016), the wake deflection due to yaw misalignment of turbines is defined by doing budget analysis on the Reynolds Averaged Navier-Stokes equations. The wake deflection angle at the rotor is defined by:

$$\theta \approx \frac{0.3\gamma}{\cos \gamma} \left( 1 - \sqrt{1 - C_T \cos \gamma} \right) \tag{16}$$

and the initial wake deflection, $\delta_0$ is defined as:

$$\delta_0 = x_0 \tan \theta \tag{17}$$

where $x_0$ is the length of the near-wake as defined in Bastankhah and Porté-Agel (2016). The total deflection of the wake as a result of wake steering is defined as:

$$\delta = \delta_0 + \frac{\theta E_0}{5.2} \sqrt{\frac{\sigma_{y0} \sigma_{z0}}{k_y k_z M_0}} \ln \left[ \frac{(1.6 + \sqrt{M_0}) \left( 1.6 \sqrt{\frac{\sigma_y \sigma_z}{\sigma_{y0} \sigma_{z0}}} - \sqrt{M_0} \right)}{(1.6 - \sqrt{M_0}) \left( 1.6 \sqrt{\frac{\sigma_y \sigma_z}{\sigma_{y0} \sigma_{z0}}} + \sqrt{M_0} \right)} \right] \tag{18}$$

where $E_0 = C_0^2 - 3e^{1/12} C_0 + 3e^{1/3}$. Expressions for the other symbols in (18) are provided in Section 2.1.3. See Bastankhah and Porté-Agel (2016) for details on the derivation.



### 2.2.3 Wake Asymmetry

Wake deflection is known to be asymmetric based on the sign of the yaw misalignment. In particular, positive yaw angles are more effective than negative yaw angles Fleming et al. (2017a). Previously, it was speculated that there was a rotation-induced lateral offset that is caused by the interaction of the wake rotation with the shear layer (Gebraad et al. (2016)). An empirical

correction used to account for asymmetry is presented in Gebraad et al. (2016).

Fleming et al. (2017a) proposes that there is an asymmetry in the wake that can be described by counter-rotating vortices, turbine rotation, and shear rather than actual deflection. Updates to the FLORIS wake modeling framework to reflect the asymmetry will be done in future work.

### 2.3 Turbine Model

In addition to wake modeling tools, a turbine model is used in the wind farm tools to provide a realistic description of turbine interactions in a wind farm. The turbine model consists of a $C_P/C_T$ table based on wind speed and constant blade pitch angle generated by a blade-element momentum code, FAST (Jonkman (2010)). The coupling between $C_P$ and $C_T$ is critical in understanding the benefits of wind farm controls. $C_P$ and $C_T$ can also be coupled using actuator disk theory, which is based on the turbine operation defined by an axial induction factor, $a$:

$$C_P = 4a(1-a)^2 \qquad (19)$$

$$C_T = 4a(1-a)$$

It is important to note that $C_P$ and $C_T$ values are used that correspond to the local conditions each turbine is operating in. For example, a turbine operating in a wake has a different $C_P/C_T$ than a turbine operating in free stream conditions.

The steady-state power of each turbine under yaw misalignment conditions is given by (Gebraad et al. (2016)):

$$P = \frac{1}{2}\rho A C_P \cos\gamma^p u^3 \qquad (20)$$

where $p$ is a tuneable parameter that matches the power loss due to yaw misalignment seen in simulations. In actuator disk theory (Burton et al. (2001)), $p = 3$. However, based on large-eddy simulations, the turbine power in yaw misalignment has been shown to match the output when $p = 1.88$ for the NREL 5 MW Jonkman et al. (2009). Field observations were run from

$p = 1.4$ (Fleming et al. (2017c)) to $p = 2.2$.

### 2.4 Lidar Model

Finally, in this work a lidar model has been added to the FLORIS wind farm framework. This lidar model is based on the University of Stuttgart's scanning lidar used in this study. This approach allows for direct comparison between lidar data collected by the scanning lidar and the wake model used. More details on the lidar used in the wake steering campaign can be

found in Raach et al. (2016); Fleming et al. (2017b); Raach et al. (2017).



The velocity computed by the scanning lidar is based on a line-of-sight velocity. The lidar model used in the FLORIS framework computes the line-of-sight velocity, $v_{LOS}$, in the same way that the lidar model computes the line-of-sight velocity so that each point that is scanned by the lidar can be compared directly to points computed by the wake model. The lidar computes a line-of-sight velocity for each point scanned. In particular, one scan point consists of $N_{weights}$ weighted points

used in a weighted sum to provide a robust velocity measurement at that scan point. The velocity at a single point can be computed as:

$$\boldsymbol{u}_i = \sum_{j=0}^{N_{weights}} a_j \tilde{\boldsymbol{u}}_{\boldsymbol{p}j} \qquad (21)$$

where $a_j$ represents the weights, which are assigned to each point, $i$ indicates the scan point, and $\boldsymbol{u} = [u_i, v_i, w_i]^T$ is the weighted sum of the measured velocity points, $\tilde{\boldsymbol{u}}_p$. Typically, the weights are assigned in a Gaussian manner.

Furthermore, the wind vector, $\boldsymbol{u} = [u_i, v_i, w_i]^T$, is projected onto the normalized laser vector point $[x_i, y_i, z_i]^T$ with a focus distance of $f_i = \sqrt{x_{i,I}^2 + y_{i,I}^2 + z_{i,I}^2}$ and the resulting $v_{LOS,i}$ being:

$$v_{los,i} = \frac{x_{i,I}}{f_i} u_{i,I} + \frac{y_{i,I}}{f_i} v_{i,I} + \frac{z_{i,I}}{f_i} w_{i,I} \qquad (22)$$

Additional details are provided in Raach et al. (2016). This model can be used in conjunction with the field data to perform closed-loop wind farm controls, as performed in Raach et al. (2016). In addition, future work could use the simulated $v_{LOS}$,

which is computed using this lidar model to fill in gaps that are inevitably present in real-time lidar data.

## 3   Field Campaign

A wake steering demonstration was conducted at NREL's National Wind Technology Center (NWTC) using a utility-scale turbine. The utility-scale turbine operated at various yaw misalignment conditions and the resulting wake was continuously recorded by a nacelle-mounted lidar. The campaign started in September 2016 and concluded in April 2017. This section

describes the turbine, the meteorological tower used to record local conditions, and the lidar system mounted on the nacelle of the turbine. Details were first reported on the lidar field campaign in Fleming et al. (2017b). This paper expands the analysis and quantitatively compares the control-oriented models presented and the lidar data collected in this campaign.

### 3.1   Setup of the Field Campaign

The turbine used in this wake steering demonstration was the U.S. Department of Energy 1.5 MW GE SLE turbine operated by

NREL (Mendoza et al. (2015); Santos and van Dam (2015); Roadman and Huskey (2015)). Details of the turbine are provided in Table 1.

The meteorological (met) tower is located 161 m upstream of the turbine in the predominant wind direction and was instrumented in accordance with International Electrotechnical Commission 61400-12-1. Table 2 lists the instrumentation used on





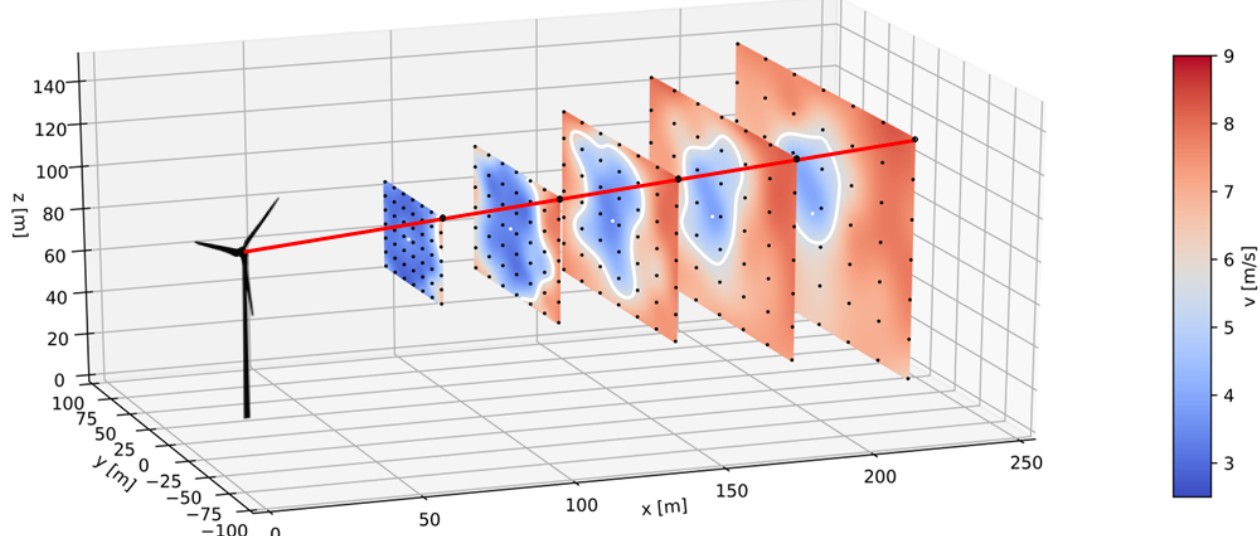

**Figure 2.** Lidar scan pattern used at the five locations downstream of the turbine.

**Table 1.** Test Turbine Details.

|  |  |
|---|---|
| Rated Power (kW) | 1500 |
| Hub Height (m) | 80 |
| Nominal Rotor Diameter (m) | 77 |
| Rated Wind Speed (m/s) | 14 |

the met tower. The turbine nacelle wind speed and wind direction are measured and recorded and synchronized with the met tower data.

**Table 2.** Meteorological Tower Instrumentation Details.

| Instrument | Elevations (m) |
|---|---|
| Precipitation | 1 |
| Wind Speed | 38, 55, 80, 87, 90, 92 |
| Wind Direction | 38, 87 |
| Humidity | 90 |
| Temperature | 38, 90 |
| Barometric Pressure | 90 |



The lidar data analyzed in this paper are limited to a region in which the met tower is upstream of the turbine. The hub-height wind speed and wind direction measurements from the met tower are used to described the mean wind speed, wind direction, and turbulence intensity. The wind direction measurements recorded at 38 m and 87 m on the met tower are used to compute

veer.

## 3.2   Lidar Specifications

The University of Stuttgart scanning lidar system consists of two parts: (1) a pulsed Windcube V1 from Leosphere and (2) a scanner unit developed at the University of Stuttgart. A two-degree of freedom mirror is used for redirecting the beam to any position within the physical limitations of the mirror. The lidar can scan an area of $(0.75D \times 0.75D)$, up to $2.8D$ downstream

using up to 49 measurement points and five scan distances. The scan rate is dependent on the number of pulses used for each measurement position. The lidar system has been used for lidar-assisted control by applying inflow and wake measurements (see Raach et al. (2016)).

The lidar scans a grid pattern shown in Figure 2 and is set to record a measurement point every 1 s on five distances from $1D$ to $2.8D$ simultaneously. Each scan consists of 49 points and takes 48 s on average. At each measurement point the lidar

uses 10,000 laser pulses to measure the line-of-sight wind speed, $v_{LOS}$, described in Section 2.4. Scans of similar atmospheric conditions and turbine operation are aggregated to produce a mean or median scan.

## 4   Results

The results presented in this section show the comparison between the wake models described in Section 2 and the lidar data collected in the field campaign described in Section 3. The results focus on comparisons of the velocity deficit behind the

turbine, the wake deflection achieved in yaw misalignment conditions, and the impact of atmospheric conditions.

### 4.1   Data Processing

It is important to note how the lidar data were processed for this study. The lidar data were first processed to filter out implausible data. Specifically, several methods were applied to check for hard target measurements, filter out lidar data with a bad carrier-to-noise ratio, and check for plausibility of the measurement data. The data is also reduced to only include: (1) periods

where the met tower is upstream of the turbine to avoid blockage effects, (2) periods where the turbine is producing at least 100 kW to capture known wake effects, and (3) periods where difference between the target and realized yaw misalignment is small.



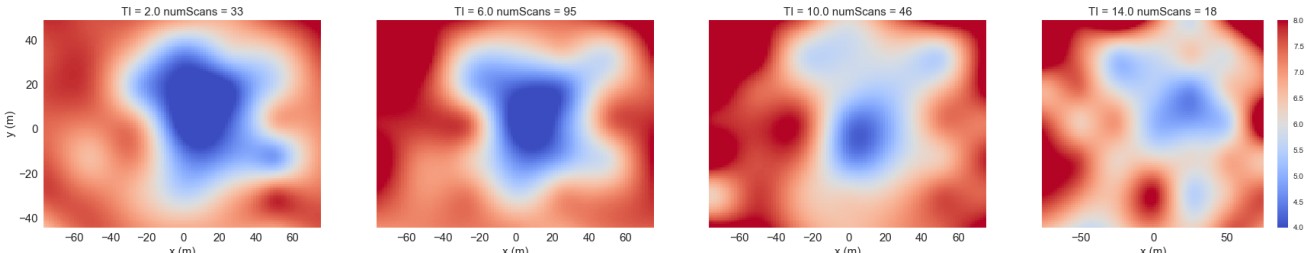

**Figure 3.** Lidar data at $2.35D$ downstream at different turbulence intensities ranging from $2.0\%$ to $14.0\%$ at 8 m/s. The title of each plot indicates the turbulence intensity and the number of scans (*numScans*) used to produce each time-averaged figure.

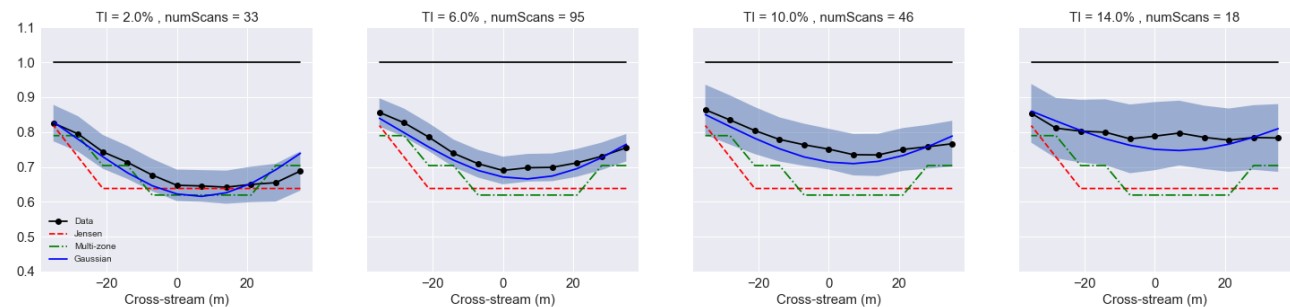

**Figure 4.** Velocity deficit at $2.35D$ downstream computed using lidar data, the Gaussian wake model, multizone wake model, and the Jensen wake model under different turbulence intensity conditions.

## 4.2 Atmospheric Conditions

First, the lidar data collected in the field campaign were analyzed based on atmospheric conditions. In particular, turbulence intensity was examined to understand the behavior of each model under varying turbulence intensity conditions. Figure 3 shows the lidar scans at $2.35D$. The turbulence intensity was computed for each lidar scan and separated into four bins with centers

5   of 2%, 6%, 10%, and 14% with a wind speed of 8 m/s. Figure 3 shows that the wake is strongest in low-turbulence conditions and dissipates quickly in high-turbulence conditions. This result is consistent with previous work investigating the effects of atmospheric conditions on wakes (Smalikho et al. (2013)).

Figure 4 shows how the controls-oriented engineering models presented in this paper compare with the lidar data. Each model was tuned to a subset of the lidar data which included primarily low-turbulence intensity data with a mean turbulence intensity

10  of approximately 5%. The velocity deficit behind the turbine was computed by averaging the velocity across a "virtual" rotor and moving this rotor across the domain in the spanwise direction (shown in Vollmer et al. (2016)). The bands indicate a 95 % confidence interval. Jensen and the multizone wake models are shown to have good agreement in low-turbulence scenarios (i.e., they fall within the confidence interval). This outcome is expected as these models were tuned to low-turbulence scenarios. However, when shifting to high turbulence intensity scenarios, the models underpredict the velocity deficit significantly. This




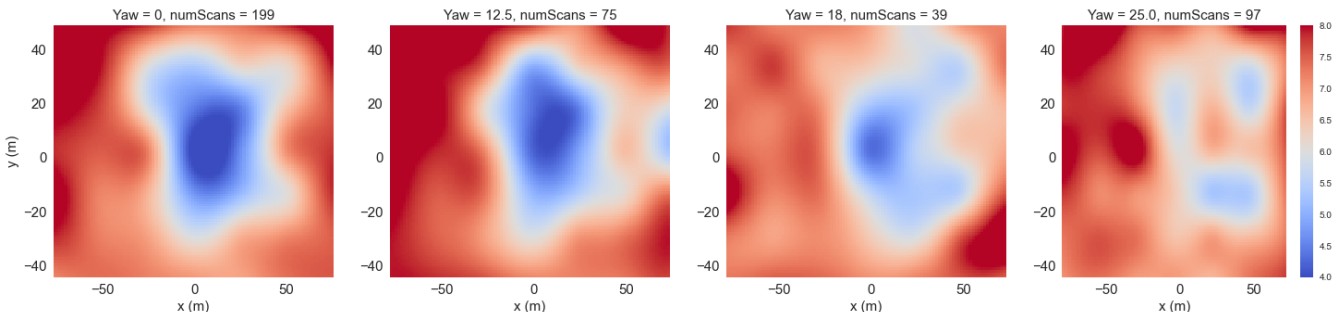

**Figure 5.** Lidar data at $2.35D$ downstream at different yaw misalignments ranging from $0°$ to $25°$ at 8 m/s. The title of each plot indicates the yaw angle and the number of scans used to produce each time-averaged figure.

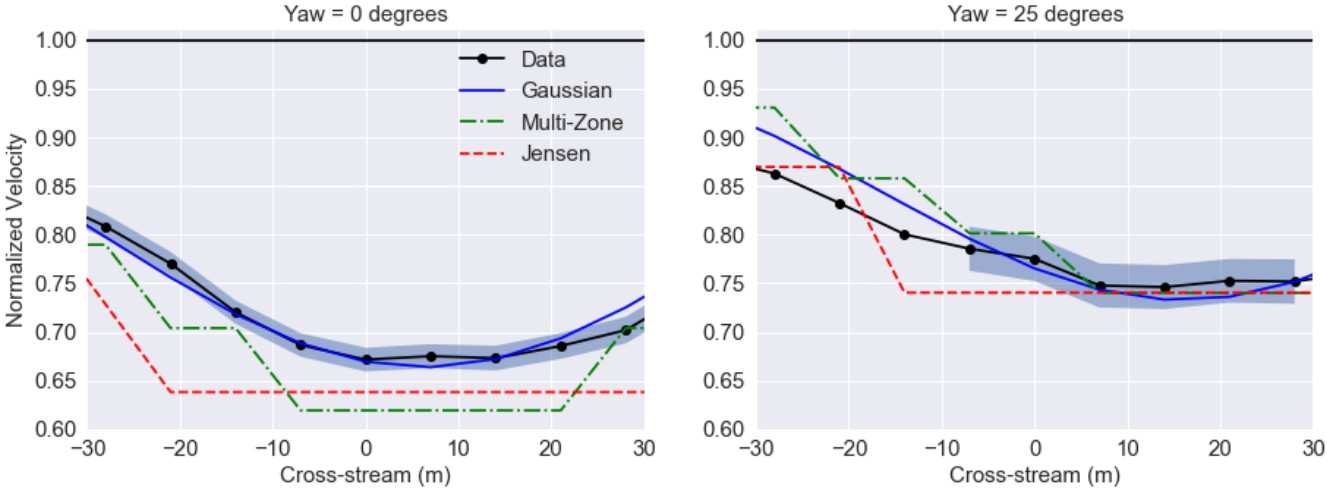

**Figure 6.** Velocity deficit at $2.35D$ downstream computed using lidar data, the Gaussian wake model, the multi-zone wake model, and the Jensen wake model under different yaw misalignment conditions.

is because neither the Jensen nor the multizone model have turbulence intensity as an input to the model. The Gaussian model, however, is able to capture both low- and high-turbulence intensity scenarios (i.e. the model lies within the confidence interval bands under each turbulence scenario examined). This capability highlights the fact that, even under varying atmospheric conditions, the Gaussian model is able to accurately capture scenarios that it was not explicitly tuned for.

## 4.3 Wake Deflection

Wake deflection was also analyzed using the lidar data from this campaign. Figure 5 shows the wake deflection under turbine yaw misalignment observed by the scanning lidar at $2.35D$ downstream. Under larger yaw angles, the wake deflects and deforms as reported in (Howland et al. (2016); Fleming et al. (2017c)).

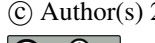



Figure 6 shows the comparison of each controls-oriented engineering model with the lidar data when the turbine is operating with no misalignment (left) and operating with $25°$ of yaw misalignment (right) at $2.35D$ downstream. Similar to Figure 4, a "virtual" rotor is used to compute the effective wind speed at several spanwise locations. The data used in Figure 6 includes all data collected in time periods when the wind speed was 7-9 m/s and the turbine was operating with no yaw misalignment and

a yaw misalignment of $25°$, regardless of turbulence intensity. The average turbulence intensity is approximately 7%. The data were aggregated and normalized over this range of wind speeds to include more scans and provide more robust statistics. The bands indicate a 95 % confidence interval.

Again, the Gaussian model is better able to predict the conditions at no misalignment (predicts velocities within the confidence intervals) since the multi-zone and Jensen model were both tuned to data with a lower turbulence intensity. When the

turbine is operating in misaligned conditions, the turbine generates a cross-flow velocity component that is not capture by the lidar. This is because the lidar is operating on a rotating platform and does not reliably measure the wake on the left side of the wake due to this large cross-flow velocity component. As a result, only lidar data from -10 m to 30 m is considered in the misaligned conditions. Under yaw misaligned conditions, the Jensen, multi-zone, and Gaussian model all have good agreement with lidar data under misaligned conditions. With more data, the analysis could be split into yaw misalignment conditions as

well as turbulence intensity levels.

## 5   Conclusions and Future Work

This paper provides a quantitative analysis of wake models used in the FLORIS framework with respect to scanning lidar data collected by NREL at the NWTC. Overall, the Gaussian wake model provides the best representation of the wake characteristics under different atmospheric conditions, specifically accounting for turbulence intensity, and different turbine operating

conditions. Good agreement was also seen with the Jensen and multizone wake model on a smaller subset of data that matched the conditions of the tuning data. Future work will include the modeling and analysis of wake deformation and changes in veer. In addition, future wind farm control strategies will be dependent on atmospheric conditions to improve their effectiveness.

*Acknowledgements.* This work was supported by the U.S. Department of Energy under Contract No. DE-AC36-08GO28308 with the National Renewable Energy Laboratory. Funding for the work was provided by the DOE Office of Energy Efficiency and Renewable Energy

Wind Energy Technologies Office. The U.S. Government retains and the publisher, by accepting the article for publication, acknowledges that the U.S. Government retains a nonexclusive, paid-up, irrevocable, worldwide license to publish or reproduce the published form of this work, or allow others to do so, for U.S. Government purposes.



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
