# Peer review of "Analysis of Control-Oriented Wake Modeling Tools Using Lidar Field Results"

_Wind Energy Science, 2018_

## Referee Comment (RC1) · Anonymous Referee #1 · 5 Apr 2018

This is an interesting paper, which compares several wake models to LIDAR measurements. The measurements are performed at a distance of $2.35D$ downstream of the turbine, and consider varying turbulence intensity and yaw angle. I believe that this paper is suitable for publication in WES, provided the authors can address the minor comments below.

1. Page 1, abstract: It would be useful for the abstract to specify that the wake models are compared to the LIDAR data at a downstream distance of $x = 2.35D$;

2. Page 2, last line: there seems to be an unnecessary parenthesis before 'Bastankhah. . .';

3. Page 4, limitations of the Jensen model: The authors should mention that, as

noted by Frandsen et al. (Wind Energ 2006; 9:39–53), the Jensen model does not conserve momentum.

4. Page 4, wake models: is there a reason to not consider the model introduced by Frandsen et al. (Wind Energ 2006; 9:39–53), which does conserve momentum? It appears that the "gaussian model" discussed here might reasonably be considered a generalization of the 'Frandsen' model. It would be interesting to test this model as well, as it might provide a way of discerning the effects of conserving momentum versus using a gaussian profile;

5. Page 5, limitations of the multizone model: are the parameters in the model constrained in a manner ensuring that the wake model conserves momentum overall?

6. Page 5, Gaussian model, before equation (6): '...is computed using a Gaussian wake based on self-similarity theory often used in free shear flows, (Pope 2000).'. A minor clarification – note that the solution shown for example in Pope (2000) is proven (not assumed) to be gaussian, based on the assumption of self-similarity. Perhaps a slight re-wording to clarify this would be '...is computed by assuming a Gaussian wake, which is inspired by self-similarity theory often used in free shear flows, (Pope 2000).'

7. Page 6, equation (9): It's not clear how this is incorporated into the model – should this affect the leading term on the right-hand-side of equation (10)?

8. Page 7, capabilities and limitations of the gaussian model: the authors should mention here that this model conserves momentum.

9. Page 13, Para. 2: the models are tuned to a subset of the data with low turbulence. What are the values of the coefficients?

10. Section 4: All LIDAR data are collected quite close to the turbine, just $2.35D$ downstream, whereas turbines are usually separated by far larger distances. Could the authors please briefly explain the value of this comparison (perhaps in the light of testing yaw models, since yaw effects are immediately discernible in the near wake)?

11. Page 14, first line: the Jensen and multizone models are said not to have turbulence intensity as an input. Could the authors nevertheless please report what values of the fitting parameters would be required to get good agreement? This would be helpful for any researchers who might wish to apply the Jensen/multizone models in high turbulence conditions.

12. Are there plans to make the data from this LIDAR campaign available, in some form, through an online repository?

---

## Referee Comment (RC2) · Anonymous Referee #2 · 17 Apr 2018

**Title:** *Analysis of Control-Oriented Wake Modeling Tools Using Lidar Field Results*

**Authors:** *J. Annoi, et al.*

**Submitted to:** Wind Energy Science Discussions

**Date:** April 12, 2018

**Reviewer general comment:** *The manuscript presents the so-called FLORIS computational platform developed for quick optimization computations related to wind farm flow dynamics and power output. Specifically, the authors develop a comparison of wake velocity deficit between the several wake models embedded in FLORS with Lidar data. This reviewer considers that it is important that the community and also industry get to know about FLORIS, and its potential advantages and short-comes. It is for this reason that I believe that this is an interesting manuscript, well written that should be published. However, before publication I would encourage the authors to consider revisiting a few 'minor' items, just with the goal to improve clarity and completeness. The most relevant issues are referenced below. Please, also note that as part of the review I introduce a commented PDF file, with a few suggestions to the wording of the document.*

**Specific Comments**

**Minor comments:**

1. **Reviewer comment:** *I had the impression that the abstract is hard to read and make a poor service to the manuscript. It wasn't until I actually read the manuscript that I understood what I was intending to read at the beginning. So please, consider clarifying the goal of the manuscript in the abstract section.*

2. **Reviewer comment:** *I did not really understood what was the reason or use of the Lidar Model in the actual manuscript? The comparisons done at the end of the manuscript are with respect to experimental data, isn't it?*

3. **Reviewer comment:** *Could you clarify the last sentence in page 12 related to the periods of data used?*

4. **Reviewer comment:** *In Figure 3 and in the later figures where the lidar data is presented, it seems that for each subfigure, corresponding to a different level of turbulence intensity, a different number of scans is used. Could the authors explain better the relationship between the number of scans and the quality of the data? Phrased differently; does the number of scans affect your results? Is the number of scans related to the availability of data, and hence it is indicated as reference for the statistical analysis of the results? Or is it related to the number of scans necessary to get a good image, dependent on turbulence conditions, and hence presence of aerosols? If the number of scans does not affect the result, maybe then the authors should present results with the same number of scans for all cases, to eliminate a free parameter? - In case this was done, is there other information that would be lost that I am not aware, or missing?*

5. **Reviewer comment:** *At the end of the results section, could the authors argue a bit more on why the Jensen and multi-zone models work better under 'medium' turbulence intensity in yawed-wake (Figure 6b) than in straight wake (Figure 6a)?*

6. **Reviewer comment:** *Please extend the discussion in the conclusions section. After making the effort of reading the manuscript, a reader would expect a more elaborated discussion/conclusion about the different wake models in FLORIS, as well as about FLORIS in general and its use for wind farm design and optimization.*

[revised manuscript text omitted]

---

## Author Comment (AC1) · 1 May 2018

[]article

[margin=1.25in]geometry

[Figure]

**Reviewer Comments**

Jennifer Annoni

We would like to thank the reviewers and editor for the time that they spent examining our manuscript. Detailed responses to all the reviewer comments are provided below. We believe the manuscript has been greatly improved by addressing the reviewer comments. Attached is the updated paper to view the changes to the manuscript.

**1  Reviewer 1**

This is an interesting paper, which compares several wake models to LIDAR measurements. The measurements are performed at a distance of 2.35D downstream of the turbine, and consider varying turbulence intensity and yaw angle. I believe that this paper is suitable for publication in WES, provided the authors address the minor comments below:

1. Page 1, abstract: It would be useful for the abstract to specify that the wake models are compared to the LIDAR data at a downstream distance of x = 2.35D.

   *A reference to the distance downstream is now mentioned in the abstract.*

2. Page 2, last line: there seems to be an unnecessary parenthesis before 'Bastankhah...'

*The parentheses were removed.*

3. Page 4, limitations of the Jensen model: The authors should mention that, as noted by Frandsen et al. (Wind Energy 2006; 9:39-53), the Jensen model does not conserve momentum.

   *This distinction has now been included in the limitations of the Jensen model and a reference to the Frandsen model is included.*

4. Page 4, wake models: is there a reason to not consider the model introduced by Frandsen et. al., which does conserve momentum? It appears that the "gaussian model" discussed here might reasonably be considered a generalization of the 'Frandsen' model. It would be interesting to test this model as well, as it might provide a way of discerning the effects of conserving momentum versus using a gaussian profile.

   *The authors agree that it would be interesting to include the Frandsen model, but in the interest of time and space, the Jensen, multi-zone, and Gaussian models were chosen. In addition, the three models chosen in this study were chosen based on their integration of wind farm controls. Each of the addressed models have been used in the context of wake steering. The authors may work to add the Frandsen model with a deflection model to future additions to the FLORIS model as another point of comparison.*

5. Page 5, limitations of the multizone model: are the parameters in the model constrained in a manner ensuring that the wake model conserves momentum overall?

   *No, momentum is not explicitly conserved. This point is now mentioned in the paper.*

6. Page 5, Gaussian model, before equation (6): '...is computed using a Gaussian wake based on self-similarity theory often used in free shear flows, (Pope 2000).'

A minor clarification - note that the solution shown for example in Pope (2000) is proven (not assumed) to be gaussian, based on the assumption of self-similarity. Perhaps a slight re-wording to clarify this would be '... is computed by assuming a Gaussian wake, which is inspired by self-similarity theory often used in free shear flows, (Pope 2000).'

*The wording has been changed as the reviewer suggested.*

7. Page 6, equation (9): It's not clear how this is incorporated into the model - should this affect the leading term on the right-hand side of equation (10)?

Equation 9 is the initialized flow field, which includes shear. The authors acknowledge the confusion and have updated equation 9 and 10 to indicate where the initial flow is incorporated. It now reads:

$$\frac{U_{init}}{U_\infty} = \left(\frac{z}{z_{hub}}\right)^{\alpha_s} \tag{1}$$

$$\frac{u(x, y, z)}{U_{init}} = 1 - Ce^{-(a(y-\delta)^2 - 2b(y-\delta)(z-z_{hub}) + c(z-z_{hub})^2)} \tag{2}$$

8. Page 7, capabilities and limitations of the gaussian model: the authors should mention here that this model conserves momentum.

*The authors include a note indicating that this model conserves momentum.*

9. Page 13, Para. 2: the models are tuned to a subset of the data with low turbulence. What are the values of the coefficients?

The following coefficients were used in this paper after being tuned and are also stated in the paper:

- Jensen

- $k_e = 0.055$
- $k_d = 0.17$

• Multi-zone

- $k_e = 0.1$
- $k_d = 0.17$
- $m_e = -0.5, 0.3, 1.0$
- $M_U = 0.47, 1.28, 5.5$
- $a_U = 11.7$
- $b_U = 0.72$

• Gaussian

- $k_a = 0.17$
- $k_b = 0.06$

10. Section 4: All LIDAR data are collected quite close to the turbine, just 2.35D downstream, whereas turbines are usually separated by far larger distances. Could the authors briefly explain the value of this comparison (perhaps in the light of testing yaw models, since yaw effects are immediately discernible in the near wake)?

*The authors acknowledge that the limitations of this field test is that the wake is measured close to the turbine. However, the authors would like to emphasize that after tuning the models to the near wake, the models are able to perform reasonably well under varying atmospheric conditions and varying turbine operations even at close proximities. In addition, as the reviewer indicates, this field test shows that wake steering can be seen at close ranges (Fleming et. al. 2017). Some work has been done to show that turbine wakes are only controllable within the first few diameters of the turbine (Raach et. al. 2016, Singh et. al. 2016). After that, the turbines become harder to control and should be described in a statistical sense.*

11. Page 14, first line: the Jensen and multizone models are said not to have turbulence intensity as an input. Could the authors nevertheless please report what values of the fitting parameters would be required to get a good agreement? This would be helpful for any researchers who might wish to apply the Jensen/multizone models in high turbulence conditions.

*The authors were able to fit the Jensen model using a value of $k_e = 0.1$ for high turbulent scenarios (TI $> 10\%$) and multizone wake model using a value of $k_e = 0.13$. The author has noted this in the Appendix section.*

12. Are there plans to make the data from this LIDAR campaign available, in some form, through an online repository?

*The data is still being processed with plans to release it to the public at a future date.*

**2   Reviewer 2**

The manuscript presents the so-called FLORIS computational platform developed for quick optimization computations related to wind farm flow dynamics and power output. Specifically, the authors develop a comparison of wake velocity deficit between the several wake models in FLORIS with Lidar data. This reviewer considers that it is important that the community and also industry get to know about FLORIS, and its potential advantages and short-comes. It is for this reason that I believe that this is an interesting manuscript, well written that should be published. However, before publication I would encourage the authors to consider revisiting a few 'minor' items, just with the goal to improve clarity and completeness. The most relevant issues are referenced below. Please, also note that as a part of the review I introduce a commented PDF file, with a few suggestions to the wording of the document.

1. I had the impression that the abstract is hard to read and make a poor service to the manuscript. It wasn't until I actually read the manuscript that I understood what I was intending to read at the beginning. So please, consider clarifying the goal of the manuscript in the abstract section.

   *The abstract has been updated to include a more clear and concise message. Below is the new abstract:*

   *The objective of this paper is to compare field data from a scanning lidar mounted on a turbine looking downstream at the turbine wake to controls-oriented wind turbine wake models. The measurements from the field campaign are used to validate controls-oriented tools used for wind plant controls and optimization. The National Wind Technology Center in Golden, CO conducted a demonstration of wake steering on a utility-scale turbine. In this campaign, the turbine was operated at various yaw misalignment set points while a lidar mounted on the nacelle scanned five downstream distances. Primarily, this paper examines measurements taken at 2.35 diameters downstream of the turbine. The lidar measurements were combined with turbine data, as well as measurements of the inflow made by a highly instrumented meteorological mast on-site. This paper presents a quantitative analysis of the lidar data as compared to the controls-oriented wake models used under different atmospheric conditions and turbine operation. These results show good agreement is obtained between the lidar data nd the models under these different conditions.*

2. I did not really understand what was the reason or the use of the Lidar model in the actual manuscript. The comparisons done at the end of the manuscript are with respect to experimental data, isn't it?

   *The lidar module was used to make sure that the 'measurements' taken from FLORIS could be compared one-to-one with the measurements in the field. Because the scanning lidar has a weighted function, some of the points are in the wake and some are outside of the wake. This provides a conservative estimate of*

*the wake and the authors wanted to be sure this was not a source of discrepancy between the models and the data. The authors acknowledge that this point was not well documented in the current version of the paper and has been updated based on the reviewer's comments. The text added is indicated below:*

*In particular, the scanning lidar used in the field campaign takes a weighted average of nine points along the line-of-sight trajectory. A lidar model is necessary to ensure this direct comparison. If any of the nine points are outside of the wake, the weighted average may lead to a more conservative estimate of the flow in the wake.*

3. Could you clarify the last sentence in page 12 related to the periods of data used?

   *The authors added a few sentences to make clear which periods of data were used in this analysis:*

   *In particular, the instruments on the met tower that are used to measure wind speed and direction are more reliable when they are not operating in the wake of nearby turbines or in the wake of its own tower due to blockage effects. We also chose to only include data where the turbine is operating normally. In this case, we define that as producing more than 100 kW. The turbine operation affects the wake properties and we need to ensure that we are comparing times when the turbine is performing as expected. Similarly, we only include times when the turbine yaw controller is tracking the specified offset within a few degrees to make a direct comparison with models.*

4. In Figure 3 and in later figures where the lidar data is presented, it seems that for each sub-figure, corresponding to different level of turbulence intensity, a different number of scans used. Could the authors explain better the relationship between the number of scans and the quality of data? Phrased differently; does the number of scans affect your results? Is the number of scans related to the availability of data, and hence it is indicated as reference for the statistical analysis of the results? Or is it related to the number of scans necessary to get a good image, dependent on turbulence conditions, and hence presence of aerosols? If the number of scans does not affect the result, maybe then the authors should present results with the same number of scans for all cases, to eliminate a free parameter? In case this was done, is there other information that would be lost that I am not aware, or missing?

*Figure 3 demonstrates the differences in the flow based on the turbulence intensity. As the reviewer indicated, the number of scans affects the quality of the data represented in Figure 3. Due to the variability of this field test, the authors chose to use the most scans as possible to gain an understanding of the general trend of the data. The authors also saw the same general trends with the smallest number of scans; however, with fewer scans, the results are more uncertain. As a result, the authors chose to keep Figure 3 as is with additional explanation in the text:*

*It is important to note that each image was generated with the maximum number of scans available after processing the data. More scans lead to a more robust measurement of the wake. A statistical analysis is presented in Figure 4, which indicates the effects of the limited number of scans processed.*

5. At the end of the results section, could the authors argue a bit more on why the Jensen and multi-zone models work better under 'medium' turbulence intensity in yawed-conditions (Figure 6b) than in straight wake (Figure 6a)?

*The authors note this is odd and could potentially be explained by the way the yaw angle modifies the thrust generated by the turbine:*

*In this case, the Jensen and multi-zone wake models have better agreement under yawed conditions than under normal conditions. One potential reason for this is that the "depth" of the wake is modified by the changing yaw angle, i.e. the thrust generated by the turbine is modified. This modified thrust is able to*

*accommodate the under predictions in the normal operating case.*

6. Please extend the discussion in the conclusions section. After making the effort of reading the manuscript, a reader would expect a more elaborated discussion/conclusion about the different wake models in FLORIS, as well as about FLORIS in general and its use for wind farm design and optimization.

*The authors have updated the discussion and conclusions sections to include a more thorough discussion of the results presented in the paper:*

[revised manuscript text omitted]

---

## Author Response (AR2)

Response to Reviewer #2:

Thank you for taking the time to re-read our manuscript.  This paper has been greatly improved throughout this process.

*Thank you for addressing my earlier comments.  At this point, I would only suggest the authors/co-authors to read once more the manuscript in search of a few left grammatical/orthography mistakes.  Also, please double check the affiliation of all authors and co-authors, I am not sure it is all correct.*

We have read through the manuscript and did our best to catch these grammatical mistakes.  We also checked the affiliations.  These are the affiliations of the people at the time of this paper.  One of the authors (Adcock) is now at Stanford.